# Application of Aquaporins as Markers in Forensic Pathology: A Systematic Review of the Literature

**DOI:** 10.3390/ijms25052664

**Published:** 2024-02-25

**Authors:** Letizia Alfieri, Angelo Montana, Paolo Frisoni, Stefano D’Errico, Margherita Neri

**Affiliations:** 1Department of Medical Sciences, University of Ferrara, 44121 Ferrara, Italy; letizia.alfieri@unife.it; 2Department of Biomedical Sciences and Public Health, University Politecnica delle Marche, 60126 Ancona, Italy; angelo.montana@ospedaliriuniti.marche.it; 3Unit of Legal Medicine, AUSL Romagna, G.B. Morgagni-L. Pierantoni Hospital, 47100 Forlì, Italy; paolo.frisoni@unife.it; 4Department of Medical Surgical and Health Sciences, University of Trieste, 34149 Trieste, Italy; sderrico@units.it

**Keywords:** aquaporins, immunohistochemical markers, forensic pathology

## Abstract

The study of aquaporins (AQPs) in various forensic fields has offered a promising horizon in response to the need to have reliable elements for the identification of the manner of death and for the individuation of forensic markers for the timing of lesions and vitality of injury. In the literature, various tissues have been studied; the most investigated are the lungs, brain, kidneys, skin, and blood vessels. A systematic literature review on PubMed following PRISMA 2020 guidelines enabled the identification of 96 articles. In all, 34 of these were enrolled to identify Aquaporin-like (AQP-like) forensic markers. The analysis of the literature demonstrated that the most significant markers among the AQPs are as follows: for the brain, AQP4, which is very important in brain trauma and hypoxic damage; AQP3 in the skin lesions caused by various mechanisms; and AQP5 in the diagnosis of drowning. Other applications are in organ damage due to drug abuse and thrombus dating. The focus of this review is to collect all the data present in the literature about the forensic application of AQPs as forensic markers in the most important fields of application. In the current use, the individuation, validation, and application of markers in forensic investigation are very useful in real forensic applications in cases evaluated in court.

## 1. Introduction

Aquaporins (AQPs) are a family of water channels/proteins and transmembrane proteins expressed in the tissues of various organs [1]. In the body, AQPs play an important role in water transport and metabolism. The first AQP described was on human erythrocyte membranes in the 1980s [2]. AQPs are organized to form tetramers, placed on the cell membrane, where they organize to form a central pore through which water, glycerol, ions, or other substances can pass depending on the subtype of AQPs.

In the literature, 17 mammalian AQPs have been identified; in particular, AQP0 to 12 have been found in humans, whereas AQP13 to 16 have been described in older lineages.

Based on the functions of aquaporins and their localization in various organs, a review of the literature was carried out based on the forensic applications of aquaporins as markers of the manner of death, vitality of lesions, or timing of injury. To use aquaporins as markers in the forensic field, it is essential to briefly clarify the functions of aquaporins in human tissues.

Thirteen types of AQPs are now known and are localized in numerous human organs like the brain, kidneys, lungs, liver, gastrointestinal tract, etc. The analysis of literature data about AQPs shows the role that these proteins play in secretion and water absorption, equilibrium between intra- and extracellular water, and angiogenesis migration and cell proliferation [3,4].

The aquaporin family is ubiquitously present in all mammalian tissues and is possibly divided into three subgroups according to its functions [1,2,4]:

(1) Aquaporins (AQP0, 1, 2, 4, 5, 6, 8) are considered a family of proteins with “pure” selective channels for the passage of water. Among these, AQP4 has been the most studied since 1994, when, for the first time, the presence of its mRNA was described in the supraoptic nuclei and the peri-vascular regions of the human brain [5].

In the central nervous system, the presence of three different aquaporins has been demonstrated, with different localizations and different functions: AQP4 is expressed in astrocytes, particularly in the cerebral cortex, and in ependymal cells [6,7]. AQP1 is expressed in the epithelial cells of the choroid plexus, and AQP9 is expressed in the endothelial cells of the subdural vessels.

AQP4 represents the most widespread bidirectional water channel, present in all structures of the central nervous system, placed in contact with the vascular compartment, and is therefore implicated in the formation and resolution of cerebral edema and the clearance of K^+^ ions released during neuronal activity [8]. The expression of AQP4 differs, however, in different areas of the central nervous system; these data suggest the multiple physiological functions of AQP4, in addition to water homeostasis. AQP4 is also implicated in cell adhesion processes in head trauma, and it plays an important role in the process of astrocyte migration that occurs in the process of post-traumatic scar formation [9].

AQP5 is expressed in the apical membrane of glands, including those of the submucosa of the airways and lacrimal, salivary, and sweat glands. The apical membrane in which AQP5 resides is the last one that water crosses during the secretion of airway fluids, tears, saliva, and sweat. Another place where AQP5 is expressed is the lung, in type 1 pneumocytes.

An answer to the role of AQP5 in the kidney was sought starting from the observation of how AQP5 is co-localized with pendrin on the apical membranes of B-type intercalated cells in the renal cortex. Since this co-expression is a common feature of other epithelia, we proceeded, in a corollary study, to evaluate whether this rigorous association could reflect a co-regulation of the two proteins [10,11].

(2) Aqua-glyceroporins (AQP3, 7, 9, 10) contribute to the cellular diffusion of water but also of glycerol, urea, and some monocarboxylates that facilitate the diffusion of lactic acid [12]. AQP1 plays a fundamental role in the formation of cerebrospinal fluid, while AQP9 is implicated in cerebral energy metabolism [8].

(3) Super-aquaporins (AQP11 and 12) are located in the cell cytoplasm and are involved in the transport of water molecules but also intra-vesicular homeostasis and the transport of larger-volume organelles [12].

Studies have shown that AQP1, 4, and 5 are also permeable to some gases such as O_2_, CO_2_, and nitric oxide [13].

The abnormal expression of AQPs can be used as a new indicator in forensic science to investigate various aspects and mechanisms. These proteins can serve as forensic markers, with AQP5 being particularly useful in distinguishing freshwater drowning from saltwater drowning [10] and in estimating methamphetamine intoxication [14]. In the field of mechanical asphyxia, a Japanese study has shown that AQP5 expression differs between asphyxial death and sudden cardiac death caused by brain lesions, making it an essential biomarker for determining the cause of death [11]. Additionally, AQP1 and AQP4 can help predict post-burn or post-traumatic cerebral edema, and genetic mutations in AQP1 and AQP9 can indicate the risk of SIDS [15,16]. By analyzing the role of AQP4 in the central nervous system, it is evident that its expression in astrocytes plays a fundamental role in the process of cerebral edema following various medical conditions such as hypoxia/asphyxia, stroke, traumatic processes, tumors, inflammation, and metabolic alterations [17]. This makes it a valuable tool in forensic science to help diagnose and determine the cause of death. The potential applications of AQPs in forensic science are vast, and research continues to uncover new uses for these proteins. These findings show the importance of AQPs in forensic science and the significance of further research to expand our understanding of their role in pathophysiology.

AQPs are involved in several cell biology aspects, and in the field of forensic medicine, the determination of the cause of death is the most important working goal on the results of various examinations [15]. In line with some of the literature, frequently, the macroscopic examination of the body must be improved and completed with ulterior analysis. The abnormal expression of each member of the AQPs can be very useful in the pathophysiology of various injuries or causes of death and may be used as a new marker or indicator in forensic pathology. This review is focused on illustrating the literature data about the application of AQPs in forensic sciences as markers for a specific cause or manner of death in forensic cases, the timing of a kind of lesion, or the vitality of injuries [15].

Therefore, considering the proven involvement of AQPs in various organs, this review aims to find all AQP applications in forensic fields, because AQPs represent excellent immunohistochemical protein markers in drowning, traumatic brain injury, and skin lesions [10,15,18,19]. This review is focused on summarizing the actual forensic applications and proposing new applications in legal medicine.

## 2. Methods

A systematic literature review was carried out up to 28 December 2023 on PubMed, following the PRISMA (Preferred Reporting Items for Systematic Reviews and Meta-Analyses) statement’s criteria in the PRISMA 2020 guidelines (Figure 1). In the preliminary identification phase, the terms (“aquaporin” and “forensic”) or (“aquaporin” and “legal medicine”) were searched in every field without time limits. From this first research, 96 articles were excluded as they met the language exclusion criterion (“not in English”). All 91 articles satisfied the inclusion criteria. Afterward, in the screening phase, based on the reading of the abstract, only studies concerning AQP applications like forensic markers were assessed for eligibility (34 articles). Finally, after a full-text review, only the articles concerning forensic pathology or legal medicine that demonstrated the use of AQP-like markers were included in this study (Figure 1).

## 3. Results and Discussion

The literature review shows various kinds of AQPs involved in the diagnoses applied to forensic pathology. AQPs are expressed by different organs like the lungs, kidneys, skin, brain, vessels, blood, and spleen. The use of these proteins as markers, in various forensic fields of application, is very useful for forensic pathologists. The forensic applications of AQPS, shown schematically in Table 1, concern asphyxial death (such as drowning, hanging, strangulation, smothering, and choking), timing skin wounds, skin injuries (such as blunt force, sharp force, strangulation marks, thermal injury, frost erythema, and gunshot wounds), burns, SIDS/SUDC, traumatic brain injuries, brain hypoxia/hypercapnia, hypoxia–ischemic brain damage, thrombosis, methamphetamine intoxication, pulmonary damage, etc.

The articles with forensic interest included in this review are listed in Table 1.

### 3.1. Brain

Numerous pieces of scientific evidence in the literature demonstrate that some proteins belonging to the aquaporin family play an important role in the formation process of cerebral edema. AQP1, AQP4, and AQP9 have a role in the regulation of water homeostasis in the brain.

AQP1 and AQP4 have been identified as the main water channel proteins in the brain [3]. In particular, aquaporin 4 (AQP4) is the one most present in the brain structures [8] and is, therefore, with a high probability, the one most involved in the process of formation of cerebral edema [47,48].

Since both AQP1 and AQP4 are rapidly induced by osmolarity, An et al. examined their expression in the brains of autopsy cases for post-mortem differentiation between SWD and FWD [38]. AQP1 was expressed on the astrocyte foot processes and blood vessels; AQP4 was found on the astrocyte foot processes, ependymal cells, and pial surfaces. Although AQP1 expression showed no significant differences, the average value of intracerebral AQP4 astrocytes was significantly increased in FWD compared with SWD [32]. Therefore, the increased expression of intracerebral AQP4 by hypotonic water to prevent hemodilution could be a useful marker in differentiating between FWD and SWD [14].

AQP4 also plays a central role in the formation process of post-burn and post-traumatic cerebral edema, which is influenced by a synergy of cellular variations at the molecular, structural, and functional levels of the blood–brain barrier. Indeed, after a traumatic brain event, an alteration of the processes regulating the homeostasis of water and ions present in the brain has been demonstrated, which contributes significantly to the poor prognosis [49].

AQP4 plays a central role in the formation process of post-traumatic brain edema. However, whether the presence of AQP4 plays a beneficial or detrimental role appears to depend on the time point concerning trauma. Edema is frequently associated with neuroinflammation with microglial activation and astrogliosis. Upregulation of AQP4 may also contribute to the neuroinflammatory process in astrogliosis and the inactivation of microglia. However, the role played by AQP4 is still not fully understood (Figure 2). In a study about fatal brain trauma, the authors retrospectively examined brain samples in cases of death after different survival times following traumatic brain injury; the results demonstrated that AQP4 was increased in patients who survived in a period between one day and three days until seven days of survival; these data suggest an upregulation of AQP4 at 3/7 days compared to 1 day since the acute stages of the hypoxic insult. AQP4 expression is correlated with neuroinflammation and hypoxia, providing evidence of the complex role of AQP4 in blunt traumatic brain injury [23].

Data from the most recent literature encourage us to hypothesize that AQP4 could be a common denominator between edema and neuroinflammation, and they underline the importance of such studies, especially given the therapeutic potential of AQP4 modulation, which could prevent the harmful effects of edema in the sequelae of head trauma [22,23,26].

Wang et al. showed that brain edema was profound in prolonged death due to severe burns. At the gene level, expression of AQP1 and AQP4 is increased in brains following prolonged death from severe burns, indicating that they are involved in the formation of post-burn cerebral edema; that is why the detection of these proteins might be a useful procedure in forensic death investigations [14].

Sudden infant death syndrome (SIDS) and sudden unexplained death in childhood (SUDC) are described as brain dysfunction that causes hypoxic stress during sleep. The expression of AQP4 in the hippocampus in SIDS/SUDC cases is associated with SIDS. The AQP4 expression in the hippocampus is lower in infants with the rs2075575 CT/TT genotype than the CC genotype and higher in the youngest infants (≤12 weeks) [27].

Other studies about SIDS have indicated a vulnerability in the development and regulation of brain function, and the authors study the genes encoding the brain aquaporins AQP1 and AQP9. In the SIDS group, an association was found between genetic variations in the AQP1 gene and maternal smoking and between the 3xTT combination in the AQP9 gene and being found lifeless in a prone position [30].

Consistent with this, Eidahl J.M.L. et al., in their original article, examined brain water content, the brain weight/body weight ratio, and the brain weight/head circumference ratio throughout the first years of life. Furthermore, they examined the relationship between these parameters and rs2075575 in the AQP4 gene, hypothesizing that dysregulated water homeostasis may be a risk factor for SIDS, which may be reflected by increased water content in the brain [28] (Figure 2).

In a study about methamphetamine brain toxicity, the author’s findings suggest that methamphetamine may induce brain damage by increasing blood–brain barrier permeability. Wang et al. examined the gene expression of AQP4 in the brain of autopsy cases, and they found a significantly enhanced expression of AQP4 in the brain following methamphetamine intoxication. This observation indicates that methamphetamine may increase AQP4 expression, eliminating accumulated water from the extracellular spaces of the brain and also activating the self-protective system [14].

### 3.2. Lungs and Kidneys

Interest in the role of AQPs in the field of drowning and asphyxia death has been investigated by several authors (Figure 3). In particular, the greatest interest has developed around the study of the lungs and, in more recent times, the study of the kidneys [34,37,38]. The application of AQPs is related to the limited degree of the lipid bilayer; the hypothesis is that a limited degree of water permeability is due to simple diffusion across the plasma membrane. However, in some tissues, the permeability to water is much greater than what one might expect from simple diffusion, which suggests that specialized and selective channels for water are present in these membranes; these types of channels are, in fact, aquaporins [3]. The movement of water through aquaporins is driven by osmotic gradients, that is, the difference in concentration between two solutions that are on opposite sides of a semi-permeable membrane, leading to the passage of water. For example, a red blood cell immersed in a hypertonic solution (seawater) shrivels up, and in a hypotonic solution (freshwater), it swells and explodes with the ingress of water.

By evaluating the tissue distribution, we can see how AQP5 is present in glandular structures and the lungs. Convincing evidence has recently been produced that AQP5 is expressed in the distal tubule of the kidney in the apical membrane of B-type intercalated cells where it co-localizes with pendrin [50].

One of the most interesting forensic applications of aquaporins lies in their application in asphyxial deaths. The diagnosis of asphyxia is one of the most difficult tasks in forensic pathology due to the absence, often, of real pathognomonic signs indicative of this manner of death, especially for smothering and choking. Therefore, various procedures have been developed to identify and explain the pathophysiology of death due to asphyxiation. Extremely relevant in this sense is a Japanese study that investigated the different pulmonary expressions of AQP1 and AQP5 in asphyxiated death compared to sudden cardiac death caused by brain lesions [33]. Indeed, a significant difference in AQP5 expression, but not in AQP1 expression, was found between smothering and choking cases and the other causes of death considered in the study (strangulation, acute cardiac and brain injury death). Immunohistochemical investigations confirmed the suppressed expression of AQP5 mRNA. Instead, AQP5-positive aggregates and granular fragments were mostly detected in the intra-alveolar spaces in cases of strangulation. The reduction in AQP5 expression in the lung could therefore be considered a specific biomarker to discriminate asphyxiated death from cardiac death [10,11,36]. Even in cases of drowning, the study of aquaporins has proven to be relevant. The immunohistochemically different expression of AQP5 and AQP2 in the samples of lungs and kidneys from freshwater drowning compared to those from saltwater drowning turned out to be a valid aspect (Figure 3). Indeed, the hypo-expression of AQP5 is a potential marker in the differential diagnosis between fresh- and saltwater drowning and can therefore be used in the diagnosis of freshwater drowning. The expression of AQP5 is more localized in pneumocytes and bronchial epithelial cells in the lungs, but there is also positivity in alveolar macrophages and the cortical collection duct system for the kidney [10,35]. On the contrary, while AQP2 is more expressed on the apical plasma membrane of collection ducts in the kidney in saltwater drowning (SWD) compared with freshwater drowning (FWD), the positivity does not show significant changes in terms of gender, age, or post-mortem interval [37,38]. Beyond the applications in the diagnosis of asphyxia, in the study by Wang et al., cases of pulmonary alveolar damage correlated with different types of injury and different survival of the subject were analyzed. All cases showed pulmonary edema, but increased pulmonary expression of AQP1, and not of AQP5, was found in deaths caused by non-rapidly fatal sharps injuries (mean survival 3–6 h), indicating a possible increase in the reabsorption capacity of alveolar fluids. The results were confirmed by studying AQP1 mRNA whose expression levels were significantly higher in delayed sharps injury deaths. The same differences were not detected in the AQP5 mRNA study [11].

### 3.3. Skin

The use of AQPs in the study of skin lesions has been reported by various authors with different applications. A study regarding the diagnosis of death associated with fire investigated AQP1 and AQP3 expression in the skin and discussed their role in the differential diagnosis of ante- and postmortem burns. In an animal experiment, the authors demonstrated that there was no difference in AQP1 gene expression, but AQP3 expression in the antemortem burn increased significantly, making it a potential forensic marker [43]. Aquaporin 3 is expressed in epidermal keratinocytes; the stratum corneum of the epidermis does not contain keratinocytes and AQP3 channels [51]. Besides that, the aquaporin 1 and 3 expression is also noticeable in thermic damage and mechanic skin lesions (blunt force, sharp force, strangulation marks, thermal injury, gunshot wounds, and frost erythema). An increased expression of aquaporin 3 in the keratinocytes of the epidermis was described in all kinds of mechanical and trauma injuries. Aquaporin 1 does not show differences in expression between injured and uninjured skin [41,44]. Regarding age estimation, several authors showed the importance of AQP1 and 3, which may also be increased in dermal vessels and keratinocytes, respectively [51,52].

Ishida et al. performed an immunohistochemical analysis of AQP1 and AQP3 in human skin wounds, concluding that the presence of more than 300 AQP3+ cells would confirm the wound age of 5 to 10 days, which is in favor of our study [44]. Regarding the estimation of the age of injuries in fresh bodies, an interesting immunohistochemistry study of 40 skin wound samples with anti-AQP3 antibodies was carried out by Ramamurthy K. et al. [40]. This study about wound age estimation was performed on forensic skin wound samples using an immunohistochemistry reaction with antibodies against AQP3. Skin samples were chosen using hematoxylin–eosin staining and selected based on the appropriate stages of wound healing. The immunohistochemistry staining with anti-AQP3 antibodies was quantified, evaluating the expression of AQP3 in injured and uninjured skin tissues. The timing of the injury was correlated with the number of AQP3-positive cells. No differences in the expression of keratinocyte aquaporin cells were found in the various age groups and between the female and male sexes. The maximal expression of cellular AQP3 was present in the proliferative phase of wound healing; in the inflammatory and maturation phases, the reaction with anti-AQP3 antibodies was less evident. They studied the role of AQP3 in wound healing concerning its expression over different healing phases, demonstrating that (a) the immunopositive AQP3 signals were expressed more in the cells of the proliferative phase than in the inflammatory and maturation phases of wound healing; (b) there was no relation between the expression of aquaporin 3-positive cells at any stage of wound healing; (c) no significant change was noted in the numbers of AQP3-positive cells concerning mode and type of injury. The aquaporin positivity was demonstrated to be independent of the manner or type of injury and the postmortem interval. The results of the study showed that the expression of AQP3 in the cells in skin wounds was maximal between 5 and 10 days, providing a marker for determining the timing of injuries of interest to the forensic field [40].

As suggested by other authors, these markers can be useful not only for determining the productive period of the lesion but also for evaluating its vitality in decomposed bodies [53].

AQP1 and 3 are also significantly enhanced in ligature marks, especially on the keratinocytes of compressed skin and in the epidermis and dermal blood vessels [39,44]. Recently, the expression of AQP3 has been observed in frost erythema in a case of lethal hypothermia and, therefore, it could be used as a marker of vitality [41,42].

### 3.4. Vessels

AQPs are widely studied for diagnosing drowning and skin wound vitality, but other forensic applications are described in the literature. The intrathrombotic expression of AQP1 and AQP3 is very interesting. In a study, this intrathrombotic expression was found in mouse models, implying that both AQP1 and AQP3 were involved in thrombogenesis and wound healing and would be useful for the determination of thrombus age. In the study performed by Nosaka et al. [46], the authors illustrated the expression of AQP1 and AQP3 in deep vein thrombosis models in mice for the individuation of antithrombotic markers. The antibodies selected for the immunohistochemical analyses were against AQP1 and AQP3. The study was performed with ligation intervals of one to five days; in thrombus samples, positive areas of AQP1 were over 70%, a decrease of less than 50% was revealed at seven days after vessel ligation, and at twenty-one days, the decrease was more evident, 11%. The positive areas of AQP3, three days after the vessel ligation, started to appear from the peripheral part of samples, and the number of AQP3-positive cells progressively increased and reached a peak 10 days after the vessel ligation. The study demonstrated that AQP1 and AQP3 are important and useful markers for the determination of thrombus age. A thrombus age of ≥10 days is indicated by the dimension of the intrathrombotic AQP1-positive area, as large as the intrathrombotic collagen area or smaller, while a thrombus age of 10–14 days is indicated by a number > 30 of AQP3-positive cells [46].

Violent lesions of the neck show various findings, both macroscopic and microscopic; particular attention could be paid to the carotid sinus. A study on the carotid sinus in cases of violence against the neck (suicidal and accidental strangulations) showed that AQP3 is not a useful marker for relevant neck pressure [45]. The study was performed on twenty-two cases of suicidal and accidental strangulations, and carotid bifurcations were examined. The analysis based on histology showed morphological alteration of hemorrhage and immunohistochemical signs of the expression of AQP3 and other proteins (heat-shock proteins 27, 60, and 70). A comparison with a control group of cases without neck lesions did not show relevant histopathological findings implying direct trauma, and no cases showed positive aquaporin 3 staining. The results demonstrated that AQP3 is not a useful marker for relevant neck pressure [45].

## 4. Conclusions

Although the pathophysiological knowledge of aquaporins is still modest, differences in AQP expression patterns are specific to causes of death and can be considered potential biomarkers in the forensic field. Therefore, it would be desirable to conduct a combined examination of several molecules including AQPs to obtain more powerful forensic evidence.

The analysis of the literature demonstrated that the most significant markers among the AQPs are AQP4 for the brain, which is very important in brain trauma and hypoxic damage; AQP3 in the skin lesions caused by various mechanisms; and AQP5 in the diagnosis of drowning in lung and kidney samples. Other fields of application are organ damage due to drug abuse and thrombus dating.

AQP1-SNPs have a high incidence in SIDS, while the expression of the AQP5 gene in the lungs of smothering would be useful for distinguishing between smothering, choking, and sudden cardiac death.

This literature review is based mostly on experimental studies conducted for different purposes, which use samples collected experimentally on different species and which also differ in type, time, and manner of death. Therefore, necessarily, unambiguous experimental conditions cannot be identified in the evaluations examined. This is due to the presence of very limited literature about the topic.

As regards the evaluations of the brain, all the articles included forensic case studies on subjects, mostly newborns [27,30,32], subjected to autopsies. In other cases, the evaluations of the encephalon started from cases, always of a forensic nature, dying from drowning [38], traumatic encephalopathy [23], severe burns, or intoxication [14]. As regards the publications selected for the evaluation of markers for AQPs in the lungs and kidneys, all the works arise from the evaluation of forensic autopsies except for the work by S. Lee et al. [35] based on experimental animal evaluation (rats). As regards the evaluation of the presence of markers for AQPs in the study of skin lesions, many of the works taken into consideration are based on forensic practice [40,41,44] but are confirmed in experimental evaluation on animals [43] and cell cultures [51]. In the evaluation of the use of aquaporins in determining the viability of vascular lesions or thrombosis, in the literature, there are both more distinctly forensic evaluations, based on autopsy cases [45], and experimental approaches on murine material [46].

In current use, also given the concordance between the evaluations that emerged on tissue models and different case studies, having evaluated the laboratory reproducibility of the methods in different contexts and the validation of the protocols used, the present work encourages the identification and application of markers in forensic investigations, which appear considerably useful in real forensic cases evaluated in court.

## Figures and Tables

**Figure 1 ijms-25-02664-f001:**
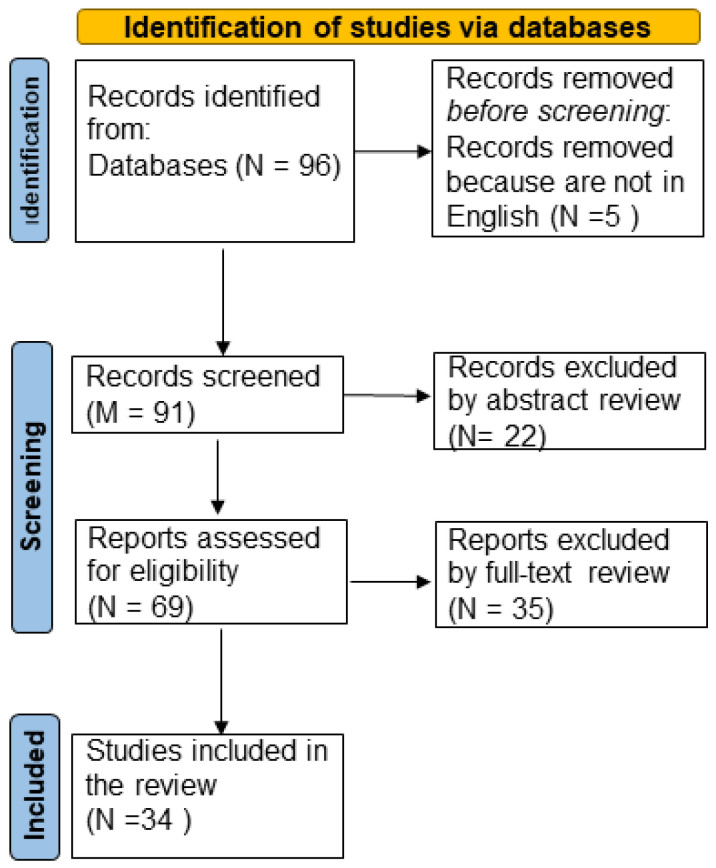
Flow diagram of study design by PRISMA 2020 guidelines.

**Figure 2 ijms-25-02664-f002:**
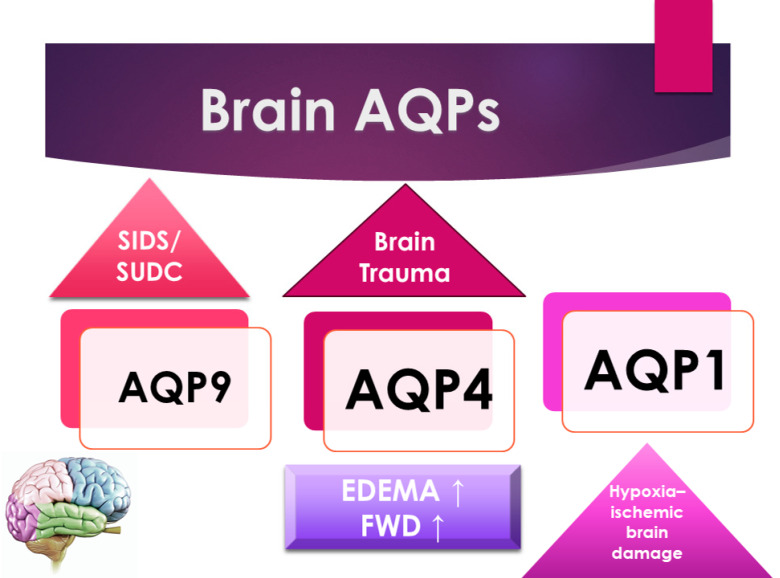
A schematic representation of the principal forensic application of AQPs in the brain.

**Figure 3 ijms-25-02664-f003:**
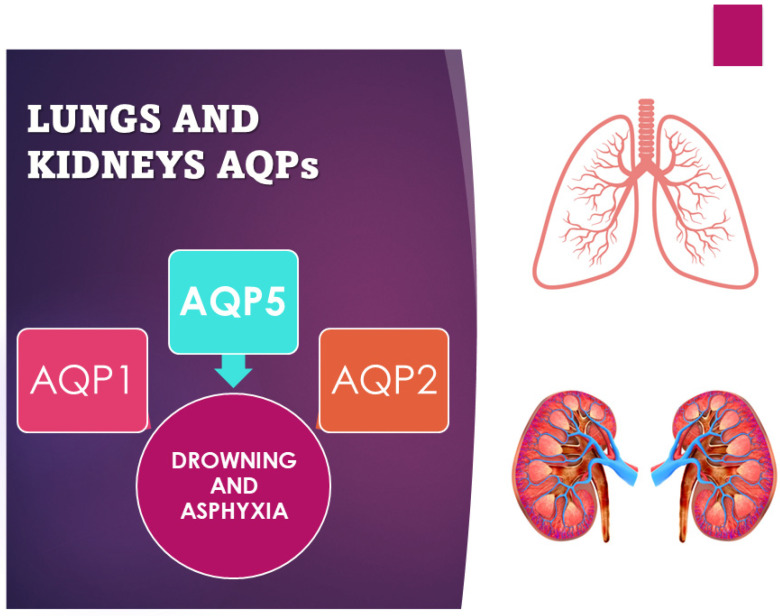
Scheme of use of AQP1, 2, and 5 as markers of asphyxia and drowning.

**Table 1 ijms-25-02664-t001:** Articles included in this review.

Samples	AQP-like Forensic Markers	Forensic Applications	Methods	Authors
**Brain**	AQP4	Brain injuries	IFC	Jiang et al., 2021 [20]
**Brain**		Traumatic brain injuries	IHC Gene Analysis and WB	Bao et al., 2012 [21] Bao et al., 2016 [22]; Neri et al., 2018 [23]; Orhan et al., 2016 [24]
**Brain**		Brain hypoxia/hypercapnia	IHC	Yu et al., 2016 [25]
**Brain**		Hypoxia–ischemic brain damage	IHC	Yu et al., 2012 [26]
**Brain (hippocampus)** **Brain**		SIDS/SUDC	IHC Gene Analysis	Eidahl et al., 2021 [27] Eidahl et al., 2023 [28]
**Brain**	AQP1 and AQP4	Methamphetamine intoxication.	IHC and Gene Analysis	Wang et al., 2014 [14]
		Drowning	IHC and IFC	An et al., 2011 [29]
**Blood and spleen**	AQP1 AQP9	SIDS	Gene Analysis	Opdal et al., 2021 [30]
**Blood and spleen**	AQP4	SIDS	Gene Analysis	Opdal et al., 2010 [31]
**Blood**	AQP4	SIDS	Gene Analysis	Opdal et al., 2017 [32]
**Lungs**	AQP1 and AQP5	Pulmonary damage	ICH and Gene Analysis	Wang et al., 2012 [11]
**Lungs**		Smothering and choking	ICH and Gene Analysis	Wang and Ishikawa et al., 2012 [33]
**Lungs**		Drowning	ICH and Gene Analysis	Hayashi et al., 2009 [34]
**Lungs**	AQP5	Drowning	PCR, WB, IHC	Lee et al., 2019 [35] Barranco et al., 2019 [36]
			IHC	
**Lungs and kidney**		Drowning	IHC	Frisoni et al., 2022 [10]
**Kidney**	AQP2	Drowning	IHC	Barranco et al., 2020 [37]
**Kidney**	AQP1, AQP2, AQP4	Drowning	IHC	An et al., 2010 [38]
**Skin**	AQP3	Strangulation	IHC	Doberentz et al., 2019 [39]
**Skin**		Timing skin wounds	IHC	Ramamurthy et al., 2023 [40]
**Skin**	AQP3 AQP1	Hanging	IHC	Prangenberg et al., 2021 [41]
		Blunt force, sharp force, strangulation marks, thermal injury, frost erythema, and gunshot wounds	IHC	Duval et al., 2020 [42]
**Skin**		Burns	IHC and Gene analysis	Kubo et al., 2013 [43]
**Review (Skin)**		Hanging	IHC	Maiese et al., 2021 [18]
**Neck skin and dermal capillaries**		Strangulation and hanging	IHC	Ishida et al., 2018 [44]
**Carotid**		Strangulation and hanging	IHC	Ulbricht et al., 2022 [45]
**Vein thrombus**		Thrombosis	IHC	Nosaka et al., 2021 [46]
**Various tissues (kidney, tongue, heart, muscle, or brain)**	AQP4	SIDS	Gene Analysis	Studer et al., 2014 [16]
**Review (various organs)**	AQP1, 2, 3, 4, 5, 6, 7, 8, 9	SIDS, drowning, skin injuries, brain injuries, intoxication	IHC, Gene Analysis	Prangenberg et al., 2021 [19]
		Drowning, wounds (vitality and timing), thermal environment, thrombus age, organ edema, Intoxication, Sudden Death	IHC, Gene Analysis	Ishida et al., 2023 [15]

## Data Availability

All the data used for the review are available from the corresponding author.

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
