# Peer review of "Application of Aquaporins as Markers in Forensic Pathology: A Systematic Review of the Literature"

_ijms, 2024, doi:10.3390/ijms25052664_

Round 1

Reviewer 1 Report

Comments and Suggestions for Authors

 I have carefully reviewed your manuscript titled "Application of Aquaporins as markers in Forensic Pathology: a systematic review of literature." Overall, the literature review on the use of aquaporins in forensic pathology is comprehensive. However, there are several issues in the presentation of the content that need attention. A well-organized review allows readers to efficiently acquire knowledge in a structured manner. The author has attempted to cover a broad range of topics but lacks logical coherence, which may pose challenges for the readers. Here are specific suggestions and concerns:

1. Compare the advantages of organizing forensic pathology based on different organs with reference to literature 17. As the cause of death is already clear and valuable from a forensic pathology perspective, what are the advantages of organizing it by organs?

2. Suggest adjusting the language in the first three paragraphs of the Introduction. For instance, combining the content of lines 39-41 and lines 33-34 might improve coherence. Similar adjustments are recommended throughout the introduction.

3. The introduction of AQPs is somewhat elaborate; a concise summary based on locations or a comprehensive introduction with organized subsections corresponding to the subsequent content is suggested.

4. During the introduction of AQPs, there is a need to strike a balance between providing background information and introducing forensic aspects, as the current content appears disjointed.

5. Ensure proper citation support for statements in lines 38-39 and 48-49.

6. Clarify the relevance of content in lines 92-93 to the subsequent discussion.

7. The formatting of lines 94-108 appears to be problematic.

8. Address redundancy in lines 119-121. Modify the expression or revise preceding content to avoid repetition.

9. Correct formatting issues in lines 141-149.

10. Table formatting needs improvement. The purpose of the table is not clear – is it focused on each AQP or on specific organs? For example, lines 157-158 mention AQP1, AQP4, and AQP9 primarily in the brain, but this is not evident in the table.

11. The two figures in the discussion section are not aesthetically pleasing and do not effectively complement the corresponding textual content.

In summary, it is my belief that the author has not identified a clear entry point, and the application of AQPs in forensic pathology is not effectively highlighted. Additionally, the presentation of the content is not reader-friendly. I recommend a thorough revision to address these concerns.

I look forward to seeing a revised version of your manuscript.

Author Response

Response to Reviewer 1

We thank Reviewer 1 for his/her evaluation of our manuscript and for helpful concerns to improve the article. In this revised version of the manuscript, we have addressed the major concerns of the referee (highlighted in yellow).

I have carefully reviewed your manuscript titled "Application of Aquaporins as markers in Forensic Pathology: a systematic review of literature." Overall, the literature review on the use of aquaporins in forensic pathology is comprehensive. However, there are several issues in the presentation of the content that need attention. A well-organized review allows readers to efficiently acquire knowledge in a structured manner. The author has attempted to cover a broad range of topics but lacks logical coherence, which may pose challenges for the readers.

Thanks again for your kind suggestions, we modified the manuscript to explain the forensic aspects better and made the paper clearer and more readable.

Here are specific suggestions and concerns:

  1. Compare the advantages of organizing forensic pathology based on different organs with reference to literature 17. As the cause of death is already clear and valuable from a forensic pathology perspective, what are the advantages of organizing it by organs?

The advantages of organizing by organs of the paper are linked to the fact that, in forensic pathology, it is not always enough to simply establish the cause of death, markers are often needed to confirm with certainty the cause of death, define the timing of an injury or establish the vitality of a lesion. From our point of view as Italian forensic pathologists, reasoning based on the functioning of the trials in Italy and the problems related to them, it seemed more useful to us to review the literature by organ rather than by type of AQPs or by cause of death.

  1. Suggest adjusting the language in the first three paragraphs of the Introduction. For instance, combining the content of lines 39-41 and lines 33-34 might improve coherence. Similar adjustments are recommended throughout the introduction.
  2. The introduction of AQPs is somewhat elaborate; a concise summary based on locations or a comprehensive introduction with organized subsections corresponding to the subsequent content is suggested.
  3. During the introduction of AQPs, there is a need to strike a balance between providing background information and introducing forensic aspects, as the current content appears disjointed.

For the points 2,3,4: we rearranged the introduction according to your suggestions, we organized the text talking in the first part about the general data of AQPs, and in the second part, we introduced the forensic aspects of the use of AQPs as markers. We rephrased some parts of the text, see the highlighted text in yellow.

  1. Ensure proper citation support for statements in lines 38-39 and 48-49.

We insert the references in the text, see the highlighted text in yellow.

  1. Clarify the relevance of content in lines 92-93 to the subsequent discussion.

The phrase is deleted because, after the revision of the introduction, it is not relevant.

  1. The formatting of lines 94-108 appears to be problematic.

We rephrase the text, see the highlighted text in yellow.

  1. Address redundancy in lines 119-121. Modify the expression or revise preceding content to avoid repetition.

We simplified the text and deleted the redundancy, see the highlighted text in yellow.

  1. Correct formatting issues in lines 141-149.

We corrected the formatting issue, see the highlighted text in yellow.

  1. Table formatting needs improvement. The purpose of the table is not clear – is it focused on each AQP or on specific organs? For example, lines 157-158 mention AQP1, AQP4, and AQP9 primarily in the brain, but this is not evident in the table.

According to your suggestion, we modified Table 1, and we have chosen to use the organs in order, listing them as they appear in the following text, see the highlighted Table in yellow.

  1. The two figures in the discussion section are not aesthetically pleasing and do not effectively complement the corresponding textual content.

First, we need to inform you that the figures were inserted following specific instructions from the Editor. We modified the figures according to your suggestion, see Figures 1 and 2 in the manuscript.

In summary, it is my belief that the author has not identified a clear entry point, and the application of AQPs in forensic pathology is not effectively highlighted. Additionally, the presentation of the content is not reader-friendly. I recommend a thorough revision to address these concerns.

We modified the abstract according to your useful suggestion, see the highlighted text in yellow.

I look forward to seeing a revised version of your manuscript.”

Thank you for your indications, we hope that the paper is now complete and clear and that we have completely satisfied your requests.

Reviewer 2 Report

Comments and Suggestions for Authors

This manuscript provides a comprehensive examination of aquaporins (AQPs) and their emerging role in forensic science, particularly in determining the cause of death, forensic diagnosis, and dating lesions. The systematic review of literature following PRISMA 2020 guidelines and the selection of 34 out of 96 articles to identify AQPs as forensic markers are commendable for their thoroughness and methodological rigor.

Two minor comments:

-since the aim of the review is to provide evidence for court, the manuscript could benefit from a more detailed discussion on the limitations of the current studies reviewed, including potential biases, the challenge of validating markers in varied forensic contexts, and the generalizability of findings;

-incorporating findings from related studies, such as the comprehensive review by Vignali G, et al. (2023) on wound vitality in decomposed bodies, could provide a broader context for the importance of AQPs in forensic science. This reference could enrich the discussion on the potential of AQPs not only as markers for dating lesions but also in assessing wound vitality, offering a novel perspective on their application in legal medicine.

Author Response

Response to Reviewer 2

We thank the Reviewer 2 for his/her evaluation of our manuscript and for helpful concerns to improve the article. In this revised version of the manuscript, we have addressed the major concerns of the referee (highlighted in green).

“This manuscript provides a comprehensive examination of aquaporins (AQPs) and their emerging role in forensic science, particularly in determining the cause of death, forensic diagnosis, and dating lesions. The systematic review of literature following PRISMA 2020 guidelines and the selection of 34 out of 96 articles to identify AQPs as forensic markers are commendable for their thoroughness and methodological rigor.

Two minor comments:

-since the aim of the review is to provide evidence for court, the manuscript could benefit from a more detailed discussion on the limitations of the current studies reviewed, including potential biases, the challenge of validating markers in varied forensic contexts, and the generalizability of findings;

-incorporating findings from related studies, such as the comprehensive review by Vignali G, et al. (2023) on wound vitality in decomposed bodies, could provide a broader context for the importance of AQPs in forensic science. This reference could enrich the discussion on the potential of AQPs not only as markers for dating lesions but also in assessing wound vitality, offering a novel perspective on their application in legal medicine.”

Thanks for the kind suggestions, very useful to improve the paper, you can see the changes inserted in the text highlighted in green.

Round 2

Reviewer 1 Report

Comments and Suggestions for Authors

The author has responded to the reviewers' comments one by one. The logic of the article is clear, the content of the table is more concise than before, and the quality of the pictures has been improved.

I have one more suggestion for the authors to consider. Since this article focuses on sorting out AQPs markers from the perspective of organs, and the content of the pictures aligns with this approach, I suggest that the author adjust the order of the table organized this time. The organs should be placed in the first column, AQPs markers in the second column, causes of death in the third column, methods in the fourth column, and references in the last column. Additionally, I believe that merging similar content would make the presentation more concise. In other words, it is not necessary for each reference to occupy a separate line; references with similar focus can be combined.

Author Response

Response to Reviewer 1

We thank Reviewer 1 for his/her evaluation of our manuscript and for helpful concerns to improve the article. In this revised version of the manuscript, we have addressed the major concerns of the referee (highlighted in light blue).

“The author has responded to the reviewers' comments one by one. The logic of the article is clear, the content of the table is more concise than before, and the quality of the pictures has been improved.

Thank you very much we are very happy that you appreciate our work and effort to improve the paper.

I have one more suggestion for the authors to consider. Since this article focuses on sorting out AQPs markers from the perspective of organs, and the content of the pictures aligns with this approach, I suggest that the author adjust the order of the table organized this time. The organs should be placed in the first column, AQPs markers in the second column, causes of death in the third column, methods in the fourth column, and references in the last column. Additionally, I believe that merging similar content would make the presentation more concise. In other words, it is not necessary for each reference to occupy a separate line; references with similar focus can be combined.”

We changed the table according to your kind suggestions, see the text highlighted in light blue in the manuscript.

Thank you for your indications, we hope that the paper is now complete and clear and that we have completely satisfied your requests.